# Oxyresveratrol Possesses DNA Damaging Activity

**DOI:** 10.3390/molecules25112577

**Published:** 2020-06-01

**Authors:** Sarayut Radapong, Satyajit D. Sarker, Kenneth J. Ritchie

**Affiliations:** 1Medicinal Plant Research Institute, Department of Medical Sciences, Ministry of Public Health, Nonthaburi 11000, Thailand; 2Centre for Natural Products Discovery, School of Pharmacy and Biomolecular Sciences, Liverpool John Moores University, Byrom Street, Liverpool L3 3AF, UK; S.Sarker@ljmu.ac.uk (S.D.S.); k.j.ritchie@ljmu.ac.uk (K.J.R.)

**Keywords:** oxyresveratrol, *Artocarpus lakoocha*, DNA damaging, pro-oxidant

## Abstract

*Artocarpus lakoocha* Wall. ex Roxb. (family: *Moraceae*) has been used as a traditional Thai medicine for the treatment of various parasitic diseases. This species has been reported to be the source of phytochemicals, which show potent biological activities. The objective of this study was to investigate the phytochemical profile of the extracts of the heartwood of *A. lakoocha* and their pro-oxidant activity in vitro. The heartwood was ground, extracted, and then chromatographic and spectroscopic analyses were carried out; oxyresveratrol was identified as the major component in the extracts. The pro-oxidant activity was investigated using DNA-nick, reactive oxygen species and reducing assays. The results showed that oxyresveratrol induced DNA damage dose-dependently in the presence of copper (II) ions. It was also found to generate reactive oxygen species (ROS) in a dose-dependent manner and reduce copper (II) to copper (I). It is concluded that oxyresveratrol is the most abundant stilbenoid in *A. lakoocha* heartwood. The compound exhibited pro-oxidant activity in the presence of copper (II) ions, which may be associated with its ability to act as an anticancer compound.

## 1. Introduction

Oxyresveratrol belongs to the group of phytochemicals known as hydroxystilbenoids and has a molecular structure similar to the well-known phytochemical resveratrol. They are monomeric stilbene comprising two aromatic rings joined by an ethylene bridge, of which, the *trans* isomer is the most common configuration (Figure 1) [1]. The natural sources of these stilbenes are abundant in grapes, itadori, peanuts, and *A. lakoocha* [2,3,4]. The compounds demonstrate similar biological activities and are used in the treatment of atherosclerosis, inflammatory, pigmentation, and carcinogenesis [5]. Interestingly, however, several biological activities are unique to oxyresveratrol (antivirus and antihelminthics) [6,7]. Further, several reports suggest that oxyresveratrol or the heartwood extract of *A. lakoocha* might cause DNA damage and cause oxidative stress [8,9,10]. Previous studies report that resveratrol and its derivatives are capable of double-stranded DNA cleavage specifically in the presence of copper ions and oxygen [10]. Piceatannol, a resveratrol derivative, fragmented DNA in human peripheral blood [9]. Some polyphenolic compounds are already well-known to cause DNA damage and are being investigated as useful leads in the development of chemotherapeutic drugs [11]. In this study, we consequently investigated the ability of oxyresveratrol from the heartwood of *A. lakoocha* to induce DNA damage and the mechanism by which the damage occurs.

## 2. Results and Discussion

### 2.1. HPLC Analysis

Phytochemical profiling of *A. lackoocha heartwood* was carried out by HPLC. Each extract was investigated (water, ethanol and ethyl acetate) and found to contain one single dominant peak with a retention time consistent with that of the chromatogramic standard of oxyresveratrol (Figure 2A). The chromatographic standard of resveratrol was also used allowing accurate calculation of the amounts of each compound in each extract (Figure 2B). Further authentication of the predominant peak was carried out using NMR with the ^13^C-NMR and ^1^H-NMR; the chemical shifts and structure elucidated in Figure 3. were in good agreement with the standard. Mass spectra of the main peak detected at RT 14.0 supposed to be oxyresveratrol showed the spectra patterns were the same as the standard (showed in the Appendix A). Its molecular formula C_14_H_12_O_4_ was deduced from the ESI-MS spectrum in positive ion mode by the hydriated molecular ion peak at *m*/*z* 245 [M + H]^+^. The amount of oxyresveratrol in the plant was consistent with the range reported by Maneechai et al. [3], while the amount of resveratrol was lower compared to the report from Borah et al. [4].

### 2.2. DNA Nicking Assay

The ability of oxyresveratrol to cause DNA damage in the presence of 50 µM copper(II) was investigated (Figure 4A). Oxyresveratrol was consequently found to cause double-strand breaks in supercoiled plasmid DNA in a dose-dependent manner (lane 5–10); 200 µM was the most effective at inducing this DNA damage, while no DNA damage was found at low oxyresveratrol concentrations (0.4–2 µM). The DNA damaging activity of oxyresveratrol was found to be dependent on the presence of Cu(II) ions, and not Fe(II) or Zn(II)(data not shown). Comparison of the DNA-damaging capacity of equivalent concentrations (50 µM) of stilbene derivatives revealed only DNA damage being induced by oxyresveratrol and resveratrol (Figure 4B), of which oxyresveratrol was the most damaging. The DNA-damaging capacity of a series of hydrostilbenoids was previously investigated and in contrast to the data reported here resveratrol was found to be the most active [6]. The DNA-damaging capacity was specifically induced by copper ions, one of the most abundant metal ions in biological systems [12]. This raises the possibility that the DNA damage observed may be due to ROS generation via the Fenton reaction consistent previous observations made by Moran et al. [13] who observed similar findings with a number of other phenolic compounds.

### 2.3. Pro-oxidant Activity of ROS Generation and Copper Reduction

Oxyresveratrol was found to generate ROS in the presence of copper(II) in a dose-dependent manner and in a higher amount compared to resveratrol and trans-stilbene (Figure 4C,D). The three extracts also exhibited ROS formation consistent with oxyresveratrol content (data not shown). Investigating the ability of oxyresveratrol to reduce Cu^2+^ ions, we report the amount of Cu^+^ produced by oxyresveratrol was dose-dependent (Figure 4E). Oxyresveratrol was also found to produce significantly higher amounts of copper (I) than resveratrol (Figure 4F). Trans-stilbene showed no copper-reducing activity; the ability of oxyresveratrol to generate ROS is consequently reported to be specifically induced by copper ions. Speculation exists as to the mechanism responsible although it has been previously noted that copper–oxo complex or copper–peroxide complex have the capacity to cause DNA damage following the reactions shown in Scheme 1 [6,14,15]. However, the exact structural feature of the complexes that caused damage to the DNA is still unclear. Copper to some certain extent is well-known to be one of the transition metals playing an important role in biological functions. However, It has been reported that significantly elevated levels of copper have been found in both serum and tissue of cancer patients [16]. Moreover, elevated copper levels have been documented in breast, cervical, ovarian, lung, prostate, stomach cancers, and leukemia. Serum copper concentration has also been found to correlate with tumour incidence and burden; malignant progression in Hodgkin’s lymphoma; and leukemic, sarcoma, brain, breast, cervical, liver, and lung cancer [17]. The generation of ROS by stilbenoids in the presence of copper may be one of the key mechanisms by which they (stilbenoids) may be useful in the treatment of cancer.

## 3. Materials and Methods

### 3.1. Standard

Oxyresveratrol (>97%), resveratrol, *trans*-stilbene, and copper acetate (Cu(OAc)_2_) (99.99%) were purchased from Sigma-Aldrich (Buchs, Switzerland); pBR322 Supercoiled plasmid DNA was purchased from Thermo Fisher Scientific (Vilnius, Lithunia); deionized water (<18 MΩ cm resistivity) was obtained from the Milli-Q Element water purification system Millipore S.A.S. (Molcheim, France).

### 3.2. Plant Material

The heartwood of *A. lakoocha* was collected from Chanthaburi province, Thailand, in June 2018. The voucher specimen (DMSC 5237) was deposited at the DMSC International Herbarium, Department of Medical Sciences, Ministry of Public Health.

### 3.3. Extraction

The heartwood was ground into powder and dried until constant weight. The dried powder was refluxed with distilled water or ethanol (250 g of the powder: 3 L of the solvents) following the procedure modified from Borah et al. [4]. The other fraction was extracted by ethyl acetate using Soxhlet apparatus (800 g of the powder: 4.5 L of the solvents). All solvents were then filtered with Whatman™ filter paper. Finally, the three extracts were concentrated and dried by rotary evaporation or lyophilizaion.

### 3.4. Chemical Analysis

Analytical HPLC was performed on a Dionex UPLC 3000 (Thermoscientific, UK) HPLC coupled with a photo-diode-array (PDA) detector (Thermoscientific). Extracts were diluted in methanol and analysed on a Phenomenex C18 column (4.6 mm × 15 cm, 5 μm, Phenomenex, Torrance, CA, USA), flow rate 1 mL/min, mobile phase gradient of water (A) and acetonitrile (B) both containing 0.1% TFA: 10–50% B, 0–30 min; 100% B, 30–32 min; 10–100% B, 33–35 min, monitored at variable UV–vis wavelengths (210, 254, 280 and 320 nm). The column temperature was set at 25 °C. The NMR spectroscopic analysis was performed in acetone-d_6_ solution on a Bruker AMX300 NMR spectrometer (300 MHz for ^13^C and ^1^H).

### 3.5. DNA Nicking Assay

The DNA break assay was modified from Subramanian et al. [10] using the reaction mixture of pBR322 plasmid DNA (200 ng) in the presence of Cu(OAc)_2_ (50 µM) or 50 µM FeCl_3_ or 50 µM ZnCl_2_ in 10 mM HEPES buffer pH 7.2. The stock solutions of oxyresveratrol, resveratrol and *trans*-stilbenoid were prepared at the concentration of 1000 µM (1mM) in 1% DMSO/Milli-Q water. Whereas, *A. lakoocha* heartwood extracts were prepared at the concentration of 1.00 mg/mL in 1% DMSO/Milli-Q water. The reactions were initiated by adding 5 µL of each compound into 20 µL of the reaction mixture and incubated at 37 °C for 15 min. After the incubation, the products were loaded into 0.80% agarose gel subjected to electrophoresis at 75 V for 1.50 h. The image of the gel was quantified by Bio-Rad documentation imaging system (Hercules, CA, USA).

### 3.6. Reactive Oxygen Species Assay

2′,7′-Dichlorodihydrofluorescein diacetate (H_2_DCFDA) was used to measure ROS generated in the reactions between oxyresveratrol in the presence or absence of copper (II) ions. The reaction was initiated by adding 50 µL of 200 µM Cu(II) or HEPES buffer into each reaction of 200 µL that contained oxyresveratrol or other derivatives. After the addition, the tubes were incubated at 37 °C for 15 min. Dichlorofluorescein (DCF), the fluorescence product of the reaction, was quantitated spectrofluorometric.

### 3.7. Copper Reducing Assay

Oxyresveratrol was colorimetrically determined using the Cu(I)-stabilizing reagent bathocuproine disulfonic acid. The reaction mixture (total volume for each tested concentration 200 µL) contained 50 µL of 2,000 mM (2M) bathocuproine disulfonic acid and 100 µL of oxyresveratrol in 10 mM HEPES buffer. The reaction was initiated by adding 50 µL of 200 mM Cu(OAc)_2_ in the sample tubes and the absorbance of the mixture at 484 nm was read at the end of 5 min.

## 4. Conclusions

Oxyresveratrol was the major stilbenoid found in all three extracts tested (water, ethanol and ethyl acetate). The compound exhibited both dose-dependent DNA damage in the presence of copper ions and ROS generation. In consideration of these findings, it can be concluded that oxyresveratrol has the potential to be developed as a promising anticancer drug candidate.

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
