# Peer review of "Oxyresveratrol Possesses DNA Damaging Activity"

_molecules, 2020, doi:10.3390/molecules25112577_

Round 1
Reviewer 1 Report
The manuscript described “Oxyresveratrol possesses DNA damaging activity”. Below are some comments and suggestions for the authors.
- Introduction part suggest add more recently researches about oxyresveratrol to strong this section. And, this sentence “Further, several reports suggest that oxyresveratrol or the heartwood extract of A. lakoocha might cause DNA damage and cause oxidative stress [5].” need cite more references not only one reference in lines 30-32.
- There are several places shown “Error! Reference source not in found”. Authors should replace another reference.
- Please correct “Cu(II)(OAc)2” to “Cu(OAc)2” in lines 95, 102 and 127, and “Zn(II)Cl2” to “ZnCl2” in line 128. And, Fe(II)Cl3 is FeCl3 or FeCl2 in line 127.
- Please correct “Cu(OAc)2” to “Cu(OAc)2” in lines 145.
- Please correct “Cu2+” to “Cu2+” and “Cu+” to “Cu+”in line 78. And, correct “Cu+-Oxo” to “Cu+-Oxo” in line 83.
- Please correct “acetone-d6” to “acetone-d6” in lines 124.
- Please correct “Subramanian et al. (2004)” to “Subramanian et al. [6]” in line 126.
- Please correct “mg/ml” to “mg/mL” in this manuscript.
9. suggest show the structure of all compounds mentioned in the introduction.
Author Response
Response to Reviewer 1 Comments
Point 1: Introduction part suggest add more recently researches about oxyresveratrol to strong this section. And, this sentence “Further, several reports suggest that oxyresveratrol or the heartwood extract of A. lakoocha might cause DNA damage and cause oxidative stress [5].” need cite more references not only one reference in lines 30-32.
Response 1: The other two references have been added in the line 34.
Point 2: There are several places shown “Error! Reference source not in found”. Authors should replace another reference.
Response 2: The references sited using Endnotex9 software have been double-checked.
Point 3: Please correct “Cu(II)(OAc)2” to “Cu(OAc)2” in lines 95, 102 and 127, and “Zn(II)Cl2” to “ZnCl2” in line 128. And, Fe(II)Cl3 is FeCl3 or FeCl2 in line 127.
Response 3: All errors have been corrected (line 149, 158 and 183).
Point 4: Please correct “Cu(OAc)2” to “Cu(OAc)2” in lines 145.
Response 4: The word has been subscripted (line 201).
Point 5: Please correct “Cu2+” to “Cu2+” and “Cu+” to “Cu+”in line 78. And, correct “Cu+-Oxo” to “Cu+-Oxo” in line 83.
Response 5: The word has been superscripted (line118-119 and 123).
Point 6: Please correct “acetone-d6” to “acetone-d6” in lines 124.
Response 6: The words in line 180 has been corrected.
Point 7: Please correct “Subramanian et al. (2004)” to “Subramanian et al. [6]” in line 126.
Response 7: The words in line 182 has been corrected.
Point 8: Please correct “mg/ml” to “mg/mL” in this manuscript.
Response 8: The units in the manuscript have been changed to mg/mL.
Point 9: suggest show the structure of all compounds mentioned in the introduction.
Response 9: The chemical structures have been added in Figure 1.

Reviewer 2 Report
The paper itself is interesting but requires checking. After that it is fit for publication.
- The whole paper is not referenced. There is an error with export from reference manager. Please correct. (Lines 44 and through the paper).
- The HPLC chromatogram in figure 1 are in very low resolution. Please include higher resolution figure.
- Figure 2 please include the error bars into the graphs.
- Line 136: H2DCFDA is 2',7'-dichlorodihydrofluorescein diacetate (see eg. https://www.thermofisher.com/order/catalog/product/D399#/D399).
Author Response
Response to Reviewer 2 Comments
Point 1: The whole paper is not referenced. There is an error with export from reference manager. Please correct. (Lines 44 and through the paper).
Response 1: The references sited using Endnotex9 software have been double-checked.
Point 2: The HPLC chromatogram in figure 1 are in very low resolution. Please include higher resolution figure.
Response 2: The chromatograms (in Figure2A.) have been replaced with the higher resolution.
Point 3: Figure 2 please include the error bars into the graphs.
Response 3: The figure has been included error bars, but they are small in Figure 4(E) and Figure 4(F).
Point 4: Line 136: H2DCFDA is 2',7'-dichlorodihydrofluorescein diacetate (see eg. https://www.thermofisher.com/order/catalog/product/D399#/D399).
Response 4: The word has been corrected (line 192).

Reviewer 3 Report
This manuscript describes the identification and DNA damaging activity of oxyresveratrol in Artocarpus lakoocha.
The topic is of interest to the journal audience.
The research is well-conceived and executed.
English needs some polishing.
Conclusions are well-supported by the data.
Issues to be addressed:
- throughout the manuscript;: "ml" should be "mL"
- the intro is rather short. Please provide more details with regards to the natural sources of oxoresveratrol
- improve image quality for figure 1 A
- fix issue with referencing (Error! Reference source not found")
- line 85: "paying" should be "playing"?
- a scheme with the structures of resveratrol and oxoresveratrol should be added in the manuscript
- a scheme describing the (assumed) mechanism for ROS production as well as the nature of the Cu(+)-Oxo complex should be added
- please provide an explanation/mechanism showing how oxoresveratrol reduces Cu2+
- Figure 2: please provide standard errors for E and F as you did for A and B
- line 124: "1H" should be "1H"
- add 1H and 13C shifts for oxoresveratrol
- line 125: within this paragraph, make sure that the number are in subscript in chemical formulas
- add an ESI with 1H and 13C spectra of crude extracts
Provided that the authors successfully address the aforementioned comments, I recommend publication of this manuscript in Molecules after major revisions.
Author Response
Response to Reviewer 3 Comments
Point 1: throughout the manuscript;: "ml" should be "mL"
Response 1: All have been changed to mL.
Point 2: the intro is rather short. Please provide more details with regards to the natural sources of oxyresveratrol.
Response 2: The natural sources of the compound have been added in the first few sentences of the introduction part.
Point 3: improve image quality for figure 1 A
Response 3: The chromatograms (in Figure2A.) have been replaced with the higher resolution.
Point 4: fix issue with referencing (Error! Reference source not found")
Response 4: The references sited using Endnotex9 software have been double-checked.
Point 5: line 85: "paying" should be "playing"?
Response 5: The word has been corrected (line 126).
Point 6: a scheme with the structures of resveratrol and oxyresveratrol should be added in the manuscript
Response 6: The chemical structures have been added in Figure 1.
Point 7: a scheme describing the (assumed) mechanism for ROS production as well as the nature of the Cu(+)-Oxo complex should be added.
Response 7: The scheme has been added (line 136-145).
Point 8: please provide an explanation/mechanism showing how oxyresveratrol reduces Cu2+
Response 8: The explanation has been shown in the discussion part and under the scheme 1.
Point 9: Figure 2: please provide standard errors for E and F as you did for A and B
Response 9: The figure has been included error bars, but they are small in Figure 4(E) and Figure 4(F).
Point 10: line 124: "1H" should be "1H"
Response 10: The word has been corrected (line 180).
Point 11: add 1H and 13C shifts for oxyresveratrol
Response 11: Both NMR chemical shifts have been added in Figure 3.
Point 12: line 125: within this paragraph, make sure that the number are in subscript in chemical formulas
Response 12: All chemical formulas are in subscript (line 183).
Point 13: add an ESI with 1H and 13C spectra of crude extracts
Response 13: We have collected the main peak from the three extracts supposed to be oxyresveratrol, run the NMR and Mass spectroscopy. The NMR and Mass spectra are shown in the supplementary materials.
Round 2
Reviewer 1 Report
The manuscript described “Oxyresveratrol possesses DNA damaging activity”. Below are some comments and suggestions for the authors.
- Please correct "2',7'-dichlorodihydrofluorescein diacetate" to "2',7'-Dichlorodihydrofluorescein diacetate" in line 192.
- Please correct "150 × 4.6 mm" to "4.6 mm × 15 cm" in line 176.
- What is "DCF" in line 195.
Reviewer 3 Report
All is good now.